# The Impact of Public Reporting Schemes and Market Competition on Hospital Efficiency

**DOI:** 10.3390/healthcare9081031

**Published:** 2021-08-11

**Authors:** Ahreum Han, Keon-Hyung Lee

**Affiliations:** 1The Department of Health Care Administration, Trinity University, San Antonio, TX 78212, USA; ahan1@trinity.edu; 2Askew School of Public Administration and Policy, Florida State University, Tallahassee, FL 32306, USA

**Keywords:** public reporting, transparency, health policy, hospital efficiency, competition

## Abstract

In the wake of growing attempts to assess the validity of public reporting, much research has examined the effectiveness of public reporting regarding cost or quality of care. However, relatively little is known about whether transparency through public reporting significantly influences hospital efficiency despite its emerging expectations for providing value-based care. This study aims to identify the dynamics that transparency brought to the healthcare market regarding hospital technical efficiency, taking the role of competition into account. We compare the two public reporting schemes, All-Payer Claims Database (APCD) and Hospital Compare. Employing Data Envelopment Analysis (DEA) and a cross-sectional time-series Tobit regression analysis, we found that APCD is negatively associated with hospital technical efficiency, while hospitals facing less competition responded significantly to increasingly transparent information by enhancing their efficiency relative to hospitals in more competitive markets. We recommend that policymakers take market mechanisms into consideration jointly with the introduction of public reporting schemes in order to produce the best outcomes in healthcare.

## 1. Introduction

Transparency in the health care sector has been identified as a tool to disseminate necessary information so that stakeholders can achieve multiple aims, including cost, quality, and access [1,2,3]. Introduced by the Affordable Care Act (ACA) in 2010, public reporting of standard charges has been promulgated across states, and since then, the federal government has adhered to an adamant stance on how compliance will be monitored and enforced. The CMS Hospital Price Transparency Rule (84 FR 65524) mandated hospitals to publicize all prices online in the form of standard charges of items and services for all payers in the machine-readable file, in addition to standard charges of 300 shoppable services, effective as of 1 January 2021. CMS will further increase penalties for noncompliance with the price transparency [4].

An increasing number of government agencies, private companies, nonprofit organizations, and even individual health care organizations offer certain types of price, quality, and satisfaction data on health care providers [1]. In particular, a number of public and private payers in the U.S. mandated public reporting as part of pay-for-performance programs and value-based purchasing [5,6]. Despite the fact that the culture of healthcare transparency has been propelled, the effectiveness of public reporting of healthcare delivery has not been fully understood yet [7]. In particular, there is scant healthcare management literature that specifically investigates the impact of public reporting on hospital efficiency.

Pursuing both cost and quality goals may create complex management trade-offs regarding efficiency [8], but efficiency is an inevitable value that cannot be ignored by healthcare management [9,10,11,12,13]. Previous studies have largely highlighted single metrics of hospital performance, either cost or quality, in relation to public reporting [14]. Measures of cost or quality alone, however, capture a limited aspect of hospital performance and may mislead overall understanding [15]. Therefore, the value relative to the resources invested should be more rigorously employed to reveal the relationship between transparency and value-based hospital outcomes.

This paper brings efficiency into the transparency dialogue. More specifically, the purpose of this study is to analyze the impact of transparency strategies on hospital technical efficiency. Hospitals face growing pressure to improve efficiency and ensure the best use of limited resources as they are considered the main cost driver in healthcare [16,17]. In this sense, it is pivotal to accurately measure hospital efficiency in assessing the impact of public reporting initiatives on hospital productivity. By examining the impact of public reporting strategies on hospital technical efficiency and taking market competition into account, this research identifies the interwoven relationships among healthcare policies, markets, and organizations.

There are two public reporting schemes of interest: the state operation of all-payer claims databases (APCDs) and Hospital Compare data released by the Center for Medicare and Medicaid Services (CMS). APCDs are state-wide archives that aggregate all different kinds of payment data, such as physician and hospital files and pharmacy or dental claims, not to mention Medicare and Medicaid. More than half of the states have been operating the database either by legislatively mandated or voluntarily (see www.apcdcouncil.org/state/map (accessed on 1 May 2021)). CMS regularly releases a variety of hospital performance datasets on the Hospital Compare website. This study incorporates outpatient imaging efficiency measures, vital in understanding whether hospitals make the conscious effort to reduce wasteful use of imaging tests, such as CT scans, MRIs, and mammograms on patients under circumstances where the imaging may be deemed medically inappropriate.

We constructed a model of the effects of public reporting and competition on hospital efficiency, as presented in Figure 1. The research design draws upon the theoretical underpinnings of institutionalism and the theory of competition [12]. We postulate that the performance of each hospital (efficiency) is determined by the institutional climate (transparency) and market conditions (competition). The following subsections demonstrate each of the three pathways that would affect the efficient operations of the hospital.

### 1.1. The Impact of Transparency on Hospital Efficiency

Institutional arrangements have a significant impact on the efficiency of healthcare delivery systems [18]. There is a plausible rationale for expecting transparency to reduce uncertainty by inducing new rules of the game into the health care market [19]. Previous studies demonstrated that the public release of performance data could help hospitals improve their focused activities and to ensure accountability [20,21]. If public reporting is institutionalized to provide health care managers with comparable and objective performance data, they are likely to make efforts to improve their performance in terms of quality and cost, which may lead to efficiency gain. Technical inefficiency is attributable to managerial motivation and efforts [22], we expect that public profiling will serve as a catalyst to encourage hospital managers to seek to enhance the technical efficiency of their organizations [9,10,23]. Thus, we hypothesize that:

**Hypothesis** **1.***Public reporting initiatives would be positively related to hospital efficiency*.

### 1.2. The Impact of Competition on Hospital Efficiency

The relationship between competition and efficiency remains unclear, with mixed results reported in the literature. Some scholars found no positive correlation between the degree of market competitiveness and efficiency [24,25,26,27,28,29,30]. Ferrari (2006) found no significant association between market-oriented reform and efficiency. It was also demonstrated that the extent of competitiveness among hospitals had no marginal effect on technical efficiency [27]. Cooper et al. (2010) found evidence that market competition with fixed prices catalyzed hospital efficiency improvement. Another research project maintained that hospitals run more efficiently in less competitive markets than in more competitive markets [28].

In contrast, a line of literature expects that competitive pressures encourage hospitals to seek higher efficiency [30,31,32,33,34,35]. Using Florida hospital data, Lee et al. (2015) found that, for a hospital located in a less competitive market, its technical efficiency score was lower than that of hospitals in a more competitive market [33]. A previous study in California demonstrated that hospitals promoted efficiency after selective contracting was implemented [34]. Hospitals succeeded in shortening patients’ length of stay, a useful indicator of efficiency, after market-based reforms [30]. Mukamel et al. (2002) concluded that higher competition was linked with lower resource allocation [35]. Despite these mixed results, this research follows the traditional economics perspective that hospitals facing greater competition will take additional measures to improve their efficiency.

**Hypothesis** **2.***When hospitals are located in a more competitive market (i.e., a lower HHI), their efficiency would be higher*.

### 1.3. The Impact of Transparency and Competition on Hospital Efficiency

We acknowledge that extant studies have focused primarily on quality, not on the efficiency of care. Economics theory suggests a complementary link between transparency and market competition [36,37]. Researchers found that transparency catalyzes a positive effect of competition on hospital performance [36,37,38,39,40]. Chou et al. (2014) maintain that hospitals in more competitive markets lowered mortality rates after report cards became available online in Pennsylvania. Zhao (2016) argued that clinical quality improvement is more noticeable in more competitive markets. Gravelle and Sivey (2010) maintained that data profiling could enhance quality only with a sufficient level of competition.

Though indirect, based on previous findings, we assume that the effect of competition would rise with increased information transparency as transparency policies would drive existing hospital markets to higher levels of predictability and certainty [41]. This paper hypothesizes as follows:

**Hypothesis** **3.***In an increasingly transparent climate, hospitals facing more competitive pressures are more likely to enhance their efficiency than those in less competitive markets*.

## 2. Methods

### 2.1. Sample and Data Sources

The study sample is comprised of nonprofit and for-profit general acute care hospitals across the states from 2011 to 2017. The total study population at the hospital level was 24,205 over the period of analysis. The average number of hospitals in a year is 3458. This study included 50 states, the District of Columbia, and 1933 counties in the U.S.

This study excludes public hospitals because they usually reside inherently in a different context from their counterparts [42]. Many public hospitals primarily serve special groups, such as Native Americans, military personnel, and veterans, not the general public. A few government-owned community hospitals are located in large urban areas to protect indigent and underprivileged populations. Most community hospitals provide a considerable amount of charity care and often experience financial distress.

There are three major data sources: the American Hospital Association (AHA)’s annual hospital survey data; hospital cost reports published by the Centers for Medicare & Medicaid Services (CMS); and the individual APCDs’ websites. Other data sources include: the Hospital Compare website, the Health Resources & Services Administration (HRSA) for the percentage of the uninsured population, and the Center for Responsive Politics for measuring lobbying efforts.

### 2.2. Model Specification

To assess the influence of public reporting and competition on hospital efficiency, we adopted two-stage statistical analyses. In the first stage, this study employs DEA using DEAP 2.1 to derive technical efficiency scores of individual hospitals relative to a best practice frontier. In the second stage, a cross-sectional time-series Tobit regression analysis is employed using Stata Version 16.

Applying DEA with Tobit models to identify the determinants of efficiency can assess policy impacts that aim for performance enhancement [43]. We used the DEA to measure the relative technical efficiency of general acute care hospitals and identify an efficiency frontier. Many previous studies analyzing single input or output variables have used ratio analysis or stochastic frontier analysis [44]. Ratio analysis compares inputs and outputs, but it fails to capture trends, while stochastic frontier analysis has the potential to treat most efficient samples as outliers [43]. Challenges with these research techniques have led to the use of data envelopment analysis (DEA) for numerous studies analyzing hospital efficiency [16,28,44].

Specifically, this study adopted the input-oriented model of the DEA method because the maximization of a hospital’s output is rarely possible as the level of healthcare provision is exogenously determined by the patients’ health status [45]. In addition, the variable-return-to-scale (VRS) model is employed in an effort to measure the potential impact of varying hospital sizes, assuming a nonlinear relationship between inputs and outputs [45].

We then use a cross-sectional time-series Tobit regression analysis to examine factors that significantly influence the technical efficiency of hospitals mainly because the dependent variable falls between zero and one. In this case, it is more logical to employ the Tobit regression model rather than the OLS regression model [17]. We integrated all variables into the following regression equation. *α* indicates intercept of the regression, and $\beta_{i}$ refers to coefficients of the independent or control variables.$$\begin{array}{l} \mathrm{Techical}\;\mathrm{efficiency}\\ \;=\alpha +\beta_{1}\mathrm{APCD}+\beta_{2}\mathrm{DataAvailibility}+\beta_{3}\mathrm{Competition}+\beta_{4}\mathrm{APCD}\\ \;*\;\mathrm{Competition}+\beta_{5}\mathrm{Data}*\mathrm{Competition}+\beta_{6}\mathrm{CMI}+\beta_{7}\mathrm{Size}\\ \;+\beta_{8}\mathrm{Urban}+\beta_{9}\mathrm{Ownership}+\beta_{10}\mathrm{Teaching}+\beta_{11}\mathrm{Medicare}\\ \;+\beta_{12}\mathrm{Medicaid}+\beta_{13}\mathrm{NPMarketshre}+\beta_{14}\mathrm{Uninsure}+\beta_{15}\mathrm{Lobbying}\\ \;+\beta_{16}\mathrm{Year}+\mathrm{uv} \end{array}$$


### 2.3. Measures

#### 2.3.1. Technical Efficiency

When calculating the technical efficiency of hospitals, four outputs and five inputs are used. The four types of outputs in this study are: the total number of visits, inpatient days, surgery, and full-time equivalent (FTE) residents. The five types of inputs include: operating expenses (excluding payroll and benefits), the number of hospital beds, the number of Doctor of Medicine (MD) FTEs, registered nurse (RN) FTEs, and other FTEs excluding trainee FTEs.

#### 2.3.2. Transparent Environments

The adoption of APCDs was used as variables of primary interest to estimate the impact of policy adoption itself as well as the history of policy adoption. To make the comparison between those who have adopted and those who have not as stark as possible, only the states that formally mandated the APCDs are considered as adopted and coded as one, while voluntary efforts or strong interests are treated as zero.

The availability of hospitals’ performance data was used to capture transparency. The transparent climate was calculated as a ratio for the number of hospitals that successfully reported their performance on outpatient imaging efficiency and the total number of hospitals in a county, as some healthcare organizations fail (or are unwilling) to report their performance. If a greater number of data become publicly available, then the likeliness of hospitals becoming exposed to transparency will increase.

#### 2.3.3. Market Competition Measures

The Hirschman–Herfindahl Index (HHI) is the standard measure used by the Department of Justice and the Federal Trade Commission to estimate the extent to which the market is concentrated. The HHIs are calculated as the sum of the squared market shares of hospitals in a hospital referral region (HRR), based on the number of adjusted admissions; a lower HHI value represents a more competitive market. The magnitude of the competition is inversely associated with this variable.

HHIs in an HRR capture the degree of market concentration in a more realistic and advantageous way than simply considering geographic market areas [46]. It is logical to estimate the degree of each hospital’s market competition based on regional healthcare resources rather than their zip codes because patients are likely to be admitted to hospitals located within the shortest distance from their residence. There are a total of 306 HHRs in the U.S, and it is known that over 85% of care is delivered by providers within a respective HRR within which 80% of the U.S. population lives [47].

#### 2.3.4. Control Variables

This research study controlled for the following characteristics at the hospital level: case mix index (CMI), size, urban, ownership, teaching affiliation, and Medicare and Medicaid discharge. Adding the CMI into the model allows comparison across hospitals by normalizing quality and cost indicators for clinical severity [48]. The number of beds is proxied for hospital size and controlled based on the assumption that larger hospitals tend to be more efficient due to economies of scale [17]. The locations of hospitals were controlled because there is an urban-rural differential in healthcare cost, utilization, access, and outcome [49]. Since the strategies of nonprofit hospitals differ from their for-profit counterparts, mainly stemming from the nature of the ownership, the ownership was controlled [42]. Teaching hospitals tend to offer a more complex range of medical services than non-teaching organizations, including tertiary care services, and thus, they are more likely to attract patients with severe diagnoses [42]. The ratio of Medicaid and Medicare patients tend to have a complicated health status or a chronic disease [50].

At a county level, the market share of nonprofit hospitals and the uninsured rate were included since the performance of hospitals relies on the institutional context [51]. In particular, a higher market share of nonprofit ownership is likely to yield better outcomes through the spillover effect on quality for the for-profit entities [52]. The overall landscape of the insured population may drive hospitals, nonprofit ones in particular, to enhance efficiency so that they can serve the uninsured better. Special interest groups, such as the American Hospital Association (AHA), are likely to take advantage of specialized knowledge to frame legislation on policy adoption and competition in favor of their interests [53]. The descriptions of each variable are organized in Table 1.

## 3. Results

### 3.1. Descriptive Analysis

The results of the 2011 and 2017 DEA are presented in Table 2, shows that the average efficiency score for acute care fluctuated during this time frame. Efficient hospitals are frontiers that achieved the optimal allocation of inputs for outputs [43]. The mean efficiency of hospitals was 0.5340, ranging from 0.4808 in 2016 to 0.5674 in 2014. Whereas hospitals maintained a similar level of average efficiency in 2011 and 2017, the number of efficient hospitals has decreased over the years except for 2014. This provides evidence of deteriorated efficiency and an increased number of slack hospitals during the investigated time frame.

Table 3 presents descriptive statistics of the variables. The availability of hospitals’ outpatient imaging efficiency data falls between 75.2% and 89.5% at the state level, while 82.4% of the data were available on average. The availability seems to vary widely across the states. The mean market competition index is 0.24. As a lower competition index means higher hospital market competition, the hospital market structures are more likely to be consolidated over time. The mean CMI was 1.507, which indicates that hospitals in the study population treated a greater number of resource-intensive caseloads than they generally do. The average number of beds did not show much change throughout the examined year. Approximately 14% of hospitals are in urban areas, and more than 77% of hospitals operate for nonprofit purposes. Teaching hospitals in the sample ranged between 877 and 927.

Medicare and Medicaid discharge comprised more than 50% and less than 20%, respectively, on average out of total patient discharge. The market share of nonprofit hospitals, out of the number of beds in a county, occupies over 80%. Given that the average portion of nonprofit ownership was 77%, it is assumable that nonprofit hospitals tend to have a slightly larger number of beds than their for-profit counterparts. The uninsured ratio has decreased continuously from 17% to 10%, along with the efforts of the ACA. The lobbying amount provided by AHA subsidiaries has been on the rise since 2011.

### 3.2. Data Envelopment Analysis of Technical Efficiency Results

Slack refers to the number of input resources that must be reduced in order for a hospital to reach the efficiency frontier [54]. A hospital is considered a frontier if its efficiency score is one and all slacks are zero, while a hospital is deemed inefficient when it can produce the same output with less input. Input slacks are the main interest of this study because this it assumes that hospitals have control only over the inputs holding outputs constant when observing an optimal production. By identifying the areas of poor resource utilization, DEA allows inefficient hospitals to benchmark the frontiers [54]. Table 4 demonstrates the average input slack resources of inefficient general acute care hospitals during 2011–2017.

From the input-oriented perspective, the analysis reported that the average excess amount of operating expenses ranged from $2.59 million in 2013 to $10.85 million in 2016. The number of beds was most used efficiently. The average overused number of the full-time equivalent (FTE) of doctors was lowest at 3.51 in 2016 and highest at 10.78 in 2012. The number of the FTE of nurses tended to be more overused than that of doctors by eight times more at maximum in 2016. Inefficient hospitals could have saved a maximum of 46.13 in the number of nurses in 2011. The total number of other FTEs, needed to be reduced by 82.41 on average for a hospital to reach the efficiency frontier in 2013.

### 3.3. Cross-Sectional Time-Series Tobit Regression Analysis Results

Table 5 shows the result of the cross-sectional time-series Tobit regression analysis. The overall model fit is robust, as indicated in the Wald $\chi_{15}^{2}$(2357.03) and in its significance level (*p* = 0.0000).

The research hypothesized that greater transparency would encourage providers to find ways to operate efficiently to at least maintain their existing respective market presences [55]. In contrast, hospitals in APCD-implementing states were likely to be less efficient than those in nonadopting states. The efficiency loss of hospitals with APCDs indicates that hospitals in states that implement APCDs may have struggled to accommodate their operations to the new environment [56]. The degree of data availability was positively related to efficiency but not statistically significant.

When a hospital operated in a more concentrated area (higher HHIs), its technical efficiency score decreased by 0.0469, echoing the traditional economics perspective. This study provides evidence that hospitals are likely to enhance operational efficiency by finding ways to improve outputs while squeezing resource inputs to maintain or raise their market shares [32]. Meanwhile, the availability of efficiency data alone did not show a statistically significant association with technical efficiency.

The interwoven relationships between data availability and technical efficiency presented a statistically significant and positive correlation when competition came into play. We found that hospitals facing less competition responded significantly to increasingly transparent information by enhancing their efficiency relative to hospitals in more competitive markets, contrary to our expectations. Most control variables were found to be significant.

## 4. Discussion

The findings in the study revealed contrasting patterns of the relationships between transparent policies and technical efficiency. State operation of APCDs reported a negative association with efficiency, while the availability of medical imaging data showed a positive link to efficiency only when the market competition was taken into account. We infer that specific transparency strategies may have different policy impacts on hospital efficiency. Although they pursue similar goals through public reporting, they differ in their specific public awareness and incentive designs, which may perplex patients and providers [57].

First, the two transparency initiatives have different levels of public awareness that come from the supremacy of ownership. Hospital Compare data, managed at the federal level, is more widely acknowledged by over 4000 Medicare-certified health care providers than by APCDs [1]. Meanwhile, states’ efforts on health care transparency can be stricken by federal decisions, as the legitimacy of state-mandated reporting of claim files to APCDs has been threatened by the Supreme Court. In Vermont, for example, the Court invalidated the state’s APCD statute that mandated the reporting of health claims data from self-insured health plans because it imposed duties that are inconsistent with the central design of the Employee Retirement Income Security Act (ERISA) [58].

Second, management theory has stressed the importance of incentive schemes because differing incentive systems result in organizational behavior consequences [59]. The two transparency approaches in the study differ, especially in terms of extrinsic motivation, which is related to external motivation [59]. Public profiling governed by CMS is closely tied with financial incentives, whereas APCD is not. Medicare’s Hospital Compare data are the standardized measurements for the federal government to reimburse the health care service costs of Medicare-participating hospitals. Hospitals are highly likely to meet the criteria in the performance measurement metrics, such as quality, safety, efficiency, or patient satisfaction, following extrinsic incentives. Meanwhile, APCDs are deployed and implemented by a state at its discretion and not necessarily linked to a state’s payment system. One suggestion for improving the efficacy of APCD is to tie the state-mandated reporting to the state Medicaid payment system.

We also learned that the market structure of competitiveness can play a critical role in increasing healthcare transparency either negatively or positively, while policy alone was not significantly associated with achieving better efficiency. Although a few previous research studies stressed the benefits of competition, this study provided a contrasting result presenting that hospitals exposed to less competition are associated with higher efficiency. One plausible reason may be found in the evidence of scale efficiencies. Economies of scale are relevant in markets with high fixed costs, such as hospitals’ maintenance and capital expenditures on medical imaging equipment [60]. However, our findings do not eliminate the possible advantages of competition under the rule of transparency. We recommend that policymakers take market mechanisms into consideration jointly with the introduction of public reporting schemes in order to produce the best outcomes in healthcare [38,61]. In addition, evidence of the impact of market competition should be examined with a nuanced understanding.

Our findings should be considered in light of the study’s limitations, chief of which is that the identification of what types of inputs are attributable to efficiency changes are out of the scope of this analysis. It is hard to pinpoint from where efficiency or inefficiency was derived because efficiency scores were derived from the production functions of inputs and outputs. Furthermore, medical care is a product of coordinated work across multiple units and caregivers, so other complex determinants of efficiency may come into play [12]. We suggest that hospital managers conduct a slack analysis to specifically identify the origins of efficiency or inefficiency in their organizations. Another limitation of this work regards the missing link between managerial efficiency and patient outcome [62]. Hospital performance must be measured ultimately by patients’ health outcomes, but this study only indicates whether managerial efficiency leads to better healthcare provision. It would be useful for future research to explore how technical efficiency promotes quality of care or patient satisfaction. The last limitation is that this study made assumptions about why policy impacts diverged without substantiating these assumptions with scientific evidence. We assumed that the divergent policy impacts may have stemmed from providers’ managerial strategies or consumers’ selection pathways. Another link which future researchers should correlate with hospital performance is the mechanisms of how policies work.

## 5. Conclusions

Skeptics may argue that public reporting not only fails to help health care providers to perform better but also generates confusion instead of clarity [57,63]. However, the move toward transparency is a major trend in healthcare [57]. The U.S. government continues to refine value-based purchasing based on containing costs, reducing waste, and increasing patient satisfaction. In this context, health services providers will continue to face pressures to accomplish two goals simultaneously: increase accountability and provide care efficiently. The two goals are not at odds, if not complementary.

We strongly suggest that efficiency or productivity should be included in any standard measurement for public reporting. Different measurements of performance evaluation can create unique market pressures on hospitals to better reduce costs and improve outcomes. Public data profiling that presents the efficiency frontier provides hospitals with an opportunity for continuous improvement of their operations [64]. From a managerial perspective, benchmarking against the top frontier will help hospitals understand their competitive position in the health care industry [43]. In addition, it is the policymakers’ and researchers’ role to standardize reporting schemes and metrics in order to make information comparable and understandable for patients and providers.

## Figures and Tables

**Figure 1 healthcare-09-01031-f001:**
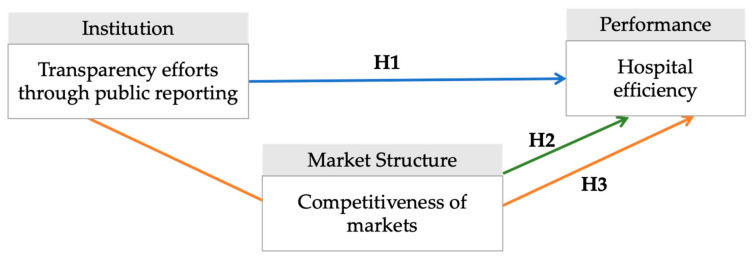
Model of the effects of public reporting and competition on efficiency.

**Table 1 healthcare-09-01031-t001:** Description of Variables.

Variables	Description
Dependent Variable	
Technical efficiency	Efficiency scores derived from the DEA method using outputs and inputs.
1. Output Variables	
(1) Outpatient visits	Total number of outpatient visits
(2) Inpatient days	Total number of inpatient days
(3) Surgeries	Total number of surgeries
(4) Residents	Total number of residents (FTEs)
2. Input Variables	
(1) Operating expenses	Total operating expenses excluding payroll and benefit
(2) Hospital beds	Total number of beds in a hospital
(3) Doctor of Medicine	Total FTEs of MDs and DOs
(4) Registered Nurse	Total FTEs of RNs
(5) Other FTEs	Total number of other FTEs excluding trainees
Independent Variables	
1. Transparent environment	
(1) APCD Adoption (1/0)	If a state adopted APCD = 1, otherwise = 0
(2) Data Availability	A ratio for the number of hospitals that successfully reported their performance and the total number of hospitals in a county (=the number of successfully reported hospitals/total number of hospitals in a county)
2. Competition (HHI)	The weighted (by hospital patient shares) hospital referral region-based HHIs for each hospital (=(each hospital’s number of adjusted admissions/total number of adjusted admissions in an HRR)^2^)
Control Variables	
1. Case Mix Index (CMI)	Case-Mix-Index from CMS
2. Size	The number of hospital beds
3. Urban	Location (urban = 1, others = 0)
4. Ownership	Hospital ownership type (Nonprofit = 1, For-profit = 0)
5. Teaching affiliation	Teaching hospital (Teaching = 1, Non-teaching = 0)
6. Medicare discharge (%)	The ratio of Medicare discharges out of total discharge
7. Medicaid discharge (%)	The ratio of Medicaid discharges out of total discharge
8. Nonprofit market share (%)	The market share of nonprofit hospitals in a county based on the total number of beds
9. Uninsured (%)	The uninsured rate of population under age 65 in a county
10. Lobbying efforts ($)	The total amount of money that AHA subsidiaries spent on lobbying efforts in a state

**Table 2 healthcare-09-01031-t002:** Summary of Efficiency Measurements.

Hospitals		2011	2012	2013	2014	2015	2016	2017
Efficient	(#)	76	69	72	63	74	55	61
	(%)	2.14%	1.98%	2.07%	1.84%	2.17%	1.61%	1.78%
	Mean	1	1	1	1	1	1	1
Inefficient	(#)	3474	3422	3414	3360	3333	3371	3361
	(%)	97.86%	98.02%	97.93%	98.16%	97.83%	98.39%	98.22%
	Mean	0.5321	0.5191	0.5187	0.5593	0.5346	0.4723	0.5375
Overall	(#)	3550	3491	3486	3423	3407	3426	3422
	Mean	0.5421	0.5286	0.5286	0.5674	0.5447	0.4808	0.5457

# = number.

**Table 3 healthcare-09-01031-t003:** Descriptive Statistics.

Variables/Year	2011	2012	2013	2014	2015	2016	2017
Observations	3550	3491	3486	3423	3407	3426	3422
Efficiency	0.5421	0.5286	0.5286	0.5674	0.5447	0.4808	0.5457
APCD (#)	16	18	20	22	22	22	23
Data Availability (%)	0.8578	0.8576	0.8596	0.8945	0.7880	0.7558	0.7524
Market Competition	0.0229	0.0237	0.0239	0.0244	0.0246	0.0246	0.0246
CMI	1.4659	1.4746	1.4833	1.4976	1.5247	1.5515	1.5583
Bed size (#)	179	181	181	181	182	182	182
Urban (#)	530	509	492	465	467	462	460
Nonprofit (#)	2742	2708	2723	2695	2686	2698	2724
Teaching affiliation (#)	878	877	883	896	927	924	915
Medicare discharge	0.5050	0.5029	0.5078	0.5131	0.5125	0.5124	0.5159
Medicaid discharge	0.1673	0.1674	0.1658	0.1774	0.1817	0.1826	0.1842
Nonprofit market share	0.8040	0.8054	0.8087	0.8097	0.8105	0.8117	0.8146
Uninsured (%)	17.16	16.81	16.61	13.41	10.88	9.95	10.20
Lobbying ($)	148,710	140,679	139,641	155,372	154,846	156,208	213,769

# = number.

**Table 4 healthcare-09-01031-t004:** Input Slack Analysis.

Slacks/Year	2011	2012	2013	2014	2015	2016	2017
Efficient hospital (#)	76	69	72	63	74	55	61
Inefficient hospital (#)	3474	3422	3414	3360	3333	3371	3361
Average Slack of Input							
Expenses ($, million)	5.04	6.75	2.59	9.02	7.52	10.85	3.4
Beds (#)	0.19	0.14	0.2	0.14	0.17	0.21	0.18
Doctor of Medicine (FTEs)	8.47	10.78	8.73	6.12	9.39	3.51	10.3
Registered Nurse (FTEs)	46.13	23.6	7.04	42.16	31.39	26.36	29.89
Other (FTEs)	43.14	38.29	82.41	28.39	39.38	42.7	27.77

**Table 5 healthcare-09-01031-t005:** Cross-sectional Time-Series Tobit Analysis Results (*N* = 24,205).

Efficiency	Coefficient		Std. Error
APCD	−0.0419	***	0.0153
Data Availability	0.0132		0.0103
HHI	−0.0469	*	0.026
APCD * HHI	0.0273		0.0346
Data Availability * HHI	0.0495	***	0.0175
CMI	0.0178	***	0.00675
Bed (#)	0.000033	**	0.000013
Urban (#)	0.0256	***	0.00617
Nonprofit (#)	0.0143	***	0.00525
Teaching affiliation (#)	0.0459	***	0.00472
Medicare discharge (%)	−0.0175	***	0.00516
Medicaid discharge (%)	0.0342	***	0.0111
Nonprofit market share (%)	0.0402	***	0.00866
Uninsured (%)	−0.00111	***	0.00033
Lobbying ($ million)	−0.0223	***	0.00629
2012	−0.0142	***	0.00208
2013	−0.0145	***	0.00211
2014	0.0182	***	0.00243
2015	−0.00563	*	0.00309
2016	−0.0701	***	0.00344
2017	−0.00435		0.00342
Constant	0.465	***	0.0169
rho	0.7097284		
Log likelihood	19,629.255		
Wald $\chi_{21}^{2}$	2357.03	***	

*** *p* < 0.01, ** *p* < 0.05, * *p* < 0.1.

## Data Availability

The data that support the findings of this study are available from the American Hospital Association, but restrictions apply to the availability of these data, which were used under license for the current study, and so are not publicly available. Data are, however, available from the authors upon reasonable request and with permission of the American Hospital Association.

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
