# Peer review of "The Impact of Public Reporting Schemes and Market Competition on Hospital Efficiency"

_healthcare, 2021, doi:10.3390/healthcare9081031_

Round 1

Reviewer 1 Report

In the abstract, include the problem and objective of the article. The methodology is transparent, but not why this research is required. 

The research problem is not clear. What is the deficiency of the previous evaluation methods that the authors propose based on DEA?
What deficiency is this new analysis able to cover?
Traditionally, there is an introduction, where a problem, objective, and hypotheses appear, and a literature review in which the gap is indicated. Here the literature review appears in a different section. I suggest integrating sections 1 and 2 into one.
Section three and four, I consider that are referring to methodology, so they should be unified. There is a total lack of scientific rigor in the organization of the article.
The methodology is very scarce. Many procedures are omitted and then appear in the results section. I doubt very much that another researcher could replicate the research reported here with the methodology presented.
Many variables are exposed in the results section that were not described in the methodology. 

Author Response

Reponses to Reviewer 1

Areas

Recommendations

Responses

Abstract

In the abstract, include the problem and objective of the article.

We revised our abstract as recommended.

Structure

I suggest integrating sections 1 and 2 into one. Section three and four, I consider that are referring to methodology, so they should be unified.

We reorganized the sections as recommended.

Research Question

The research problem is not clear.

We included the following paragraph on page 1.

“The purpose of this study is to analyze the impact of two public reporting initiatives on hospital technical efficiency: All-Payer Claims Databases (APCDs) and Hospital Compare data released by CMS. We also examine the role of competition in the healthcare market in the new institutional arrangements toward transparency.”

Contribution

The methodology is transparent, but not why this research is required. 

We added the following to paragraphs to demonstrate why this research is necessary and important on page 2.

“The effectiveness of public reporting of healthcare delivery has not been fully understood as findings are uncertain [7]. In particular, there is scant healthcare management literature which specifically investigates the impact of public reporting on hospital efficiency. Previous studies have largely highlighted single metrics of hospital performance, either cost or quality, in relation with public reporting [8]. Measures of cost or quality alone, however, capture a limited aspect of hospital performance and may mislead overall understanding [9]. Therefore, efficiency measures should be more rigorously employed to reveal the relationship between transparency and value-based outcome of hospitals.         

This paper aims to fill this research gap by bringing efficiency into public reporting discussion……”

Method

What is the deficiency of the previous evaluation methods that the authors propose based on DEA? What deficiency is this new analysis able to cover?

We added the following paragraph to deal with the deficiency of the previous evaluation methods on pages 5-6.

“We used the DEA to measure the relative technical efficiency of general acute care hospitals and identify an efficiency frontier. Many previous studies analyzing single input or output variables has used ratio analysis or stochastic frontier analysis [44]. Ratio analysis compares inputs and outputs, but it fails to capture trends, while stochastic frontier analysis has the potential to treat most efficient samples as outliers [43]. Challenges with these research techniques have led to the use of data envelopment analysis (DEA) for numerous studies analyzing hospital efficiency.

Method

The methodology is very scarce. Many procedures are omitted and then appear in the results section. I doubt very much that another researcher could replicate the research reported here with the methodology presented.

We added the description of variables (Table 1 on page 4) and more explanations on DEA methodology to help readers better understand our method and replicate if necessary.

Method

Many variables are exposed in the results section that were not described in the methodology. 

We believe the newly inserted Table 1 will help readers find what variables were used in this study and how they were measured.   

Reviewer 2 Report

This paper analyzes the relationship between public reporting schemes and hospital efficiency, and examines the mediating role of competition in this relationship. Authors need to make a few corrections to improve the quality of this paper. 

First, please supplement the discussion of the relationship between public reporting and efficiency. Also, please enrich the discussion on the mediating role of competition. For example, you can add related previous studies on public reporting, efficiency, and competition. 

Second, please improve the scientific soundness of your research design. The paper lacks a sufficient description of the research design. 

Third, please add abundant previous studies and logic to support your analysis results. In addition, add interpretation of the analysis results. 

Fourth, please add theoretical and practical implications of this paper. This manuscript lacks explanations on implications. 

Fifth, please supplement the contents of the conclusions more abundantly. In particular, suggest the limitations of this paper and ways to overcome them. 

Author Response

Reponses to Reviewer 2

Areas

Recommendations

Responses

Literature review

Supplement the discussion of the relationship between public reporting and efficiency. Also, please enrich the discussion on the mediating role of competition. For example, you can add related previous studies on public reporting, efficiency, and competition. 

We supplemented the discussion of the relationship among policy, competition, and efficiency on pages 2-3. Please find sections 1.1-1.3.

Method

Improve the scientific soundness of your research design. The paper lacks a sufficient description of the research design. 

We added the description of variables (Table 1 on page 4) and more explanations on DEA methodology to help readers better understand our method and replicate if necessary.

Results

Add abundant previous studies and logic to support your analysis results. In addition, add interpretation of the analysis results. 

We did our best to include relevant literature more.  We added our explanation mainly in the discussion part. We hope you see our analysis results are sufficiently explained.

Implication  

Add theoretical and practical implications of this paper. This manuscript lacks explanations on implications. Supplement the contents of the conclusions more abundantly.

As recommended, we bolstered our implications and suggestions for health care managers and policymakers in the conclusion section on pages 10-11.

Limitation 

Suggest the limitations of this paper and ways to overcome them.

As recommended, we added the limitations of this study on page 10.

Round 2

Reviewer 1 Report

Thanks for attending to my comments. 

Author Response

Thank you very much.

Reviewer 2 Report

Authors corrected their manuscript by addressing reviewer's comments. However, some of the reviews were not sufficiently reflected. 

First, previous studies on the mediating role of competition have
not been enough added. Although it is important to logically explain the role of competition because it is related to the research model of this paper. However, a rational basis for this has not been presented. 

Second, in response to the reviewer's comment to improve the scientific soundness of research design, the authors made a description of variables and explanations on DEA methodology. However, it is not an appropriate response.  Variables and methodologies are important, but more importantly, it is necessary to logically develop a research model and persuade readers. Authors need to add explanations based on the theoretical background of the research design. 

Third, despite the reviewer's comments, the theoretical implications of the manuscript presented by the authors are still lacking. Authors need to add how this paper contributes academically to the relevant literature. 

Theses comments may help authors to improve the contribution of their manuscript to academia and practice.

Author Response

Reponses to Reviewer 2

Areas

Recommendations

Responses

Literature

First, previous studies on the mediating role of competition have not been enough added. Although it is important to logically explain the role of competition because it is related to the research model of this paper. However, a rational basis for this has not been presented.

First of all, we changed our title of the manuscript by removing the mediating role of competition. Despite of such change, we value the reviewer’s suggestion to add more explanation on the mediating role of competition as it is still related to our findings. Thank you.

According to your recommendation, we included the following paragraph on 2-3.

Economic theory suggests a complementary link between transparency and market competition [27,28]. Researchers find that clinical quality improvement was more noticeable in a more competitive markets [27–31]. Chou and her colleagues (2014) maintain that hospitals in more competitive markets lowered mortality rate after public reporting card became available online in Pennsylvania. Gravelle and Sivey (2010) maintained that data profiling can enhance quality only with a sufficient level of competition. Zhao (2016) suggests that public reporting and market structure should be jointly considered to produce best outcomes in healthcare in the policymaking arena [11]. Based on previous findings, we assume that the effect of competition would become magnified with increased information transparency as transparency policies would drive the existing hospital markets to be more predictable and certain [32]. This paper hypothesizes that an increasingly transparent climate, hospitals facing more competitive pressures is likely to enhance their efficiency more than those in less competitive markets. 

Research Design

Variables and methodologies are important, but more importantly, it is necessary to logically develop a research model and persuade readers. Authors need to add explanations based on the theoretical background of the research design.

There are three strands of theoretical framework this study employs, and the theoretical backgrounds are developed into three hypotheses. (See pages 2-3)

1. Transparency à efficiency

Institutional arrangements have a significant impact on the efficiency in healthcare delivery systems [9]. There is a plausible rationale for expecting transparency to reduce uncertainty by inducing new rules of the game into the health care market [10]. Previous studies demonstrated that the public release of performance data could help hospitals to improve their focused activities and to ensure accountability of providers [11,12]. If public reporting is institutionalized to provide health care managers with comparable and objective performance data, they are likely to make efforts to improve their performance in terms of quality and cost, which may lead to efficiency gain. From the literature, it is regarded that managerial ability in production is related to technical efficiency [13,14]. Based on Leibenstein’s (1966) argument that technical inefficiency is attributable to a lack of managerial motivation and efforts, we expect that public profiling will serve as catalyst to encourage hospital managers to make efforts in enhancing technical efficiency [15]. Thus, we hypothesize that public reporting would be positively related to hospital efficiency.

2. Competition à efficiency

The relationship between competition and efficiency remains unclear with mixed results reported in the literature [16–18]. Some scholars found no positive correlation between the degree of market competitiveness and efficiency [16,19,20]. Ferrari (2006) found no significant association between market-oriented reform and efficiency. It was also found that the extent of competitiveness among hospitals had no marginal effect on technical efficiency [21]. In contrast, a line of literature expects that competitive pressures encourage hospitals to seek higher efficiency [17,22–24]. A previous study in California demonstrated that hospitals promoted efficiency after selective contracting was implemented [25]. Hospitals succeeded in shortening patients’ length of stay by operating more efficiently after market-based reforms [22]. Mukamel and others (2002) concluded that higher competition was linked with lower expenditures [26]. Using Florida hospital data, Lee et al. (2015) found that, for a hospital located in a less competitive market, its technical efficiency score was lower than that of hospitals in a more competitive market [24]. Despite these mixed results, this research hypothesizes that the competitiveness of the market is positively associated with hospital efficiency based on traditional economics theory.

3. Transparency & competition à efficiency

Economic theory suggests a complementary link between transparency and market competition [27,28]. Researchers find that clinical quality improvement was more noticeable in a more competitive markets [27–31]. Chou and her colleagues (2014) maintain that hospitals in more competitive markets lowered mortality rate after public reporting card became available online in Pennsylvania. Gravelle and Sivey (2010) maintained that data profiling can enhance quality only with a sufficient level of competition. However, we acknowledge that the extant studies have focused primarily on quality of care, not efficiency. Though indirect, based on the previous findings, we assume that the effect of competition would become magnified with increased information transparency as transparency policies would drive the existing hospital markets to be more predictable and certain [32]. This paper hypothesizes that an increasingly transparent climate, hospitals facing more competitive pressures is likely to enhance their efficiency more than those in less competitive markets. 

Implication  

The theoretical implications of the manuscript presented by the authors are still lacking. Authors need to add how this paper contributes academically to the relevant literature. 

This research contributes to the literature evaluating the efficiency of U.S. hospitals under the umbrella of healthcare transparency at the state and federal levels and is the first attempt to examine the interwoven relationships among healthcare policies, markets, and organizations (page 2).

We believe this study sufficiently provides and useful theoretical and policy implications. For example, based on management theory that stressed the importance of incentive schemes because differing incentive systems result in organizational behavior consequences, we explained how the two transparency efforts differ in terms of extrinsic motivation and the consequences of the different design (page 10). We additionally offered our finding in relation with traditional market competition theory and derived a practical recommendation that policymakers take market mechanisms into consideration with the introduction of public reporting schemes in order to produce the best outcomes in healthcare (page 10).